# *Citrus limon* Essential Oil: Chemical Composition and Selected Biological Properties Focusing on the Antimicrobial (In Vitro, In Situ), Antibiofilm, Insecticidal Activity and Preservative Effect against *Salmonella enterica* Inoculated in Carrot

**DOI:** 10.3390/plants13040524

**Published:** 2024-02-15

**Authors:** Miroslava Kačániová, Natália Čmiková, Nenad L. Vukovic, Andrea Verešová, Alessandro Bianchi, Stefania Garzoli, Rania Ben Saad, Anis Ben Hsouna, Zhaojun Ban, Milena D. Vukic

**Affiliations:** 1Institute of Horticulture, Faculty of Horticulture and Landscape Engineering, Slovak University of Agriculture, Tr. A. Hlinku 2, 94976 Nitra, Slovakia; n.cmikova@gmail.com (N.Č.); andrea.veresova1979@gmail.com (A.V.); milena.vukic@pmf.kg.ac.rs (M.D.V.); 2School of Medical & Health Sciences, University of Economics and Human Sciences in Warsaw, Okopowa 59, 01043 Warszawa, Poland; 3INTI International University, Persiaran Perdana BBN Putra Nilai, Nilai 71800, Malaysia; 4Department of Chemistry, University of Kragujevac, Faculty of Science, R. Domanovića 12, 34000 Kragujevac, Serbia; nvchem@yahoo.com; 5Department of Agriculture, Food and Environment, University of Pisa, Via del Borghetto 80, 56124 Pisa, Italy; alessandro.bianchi@phd.unipi.it; 6Department of Chemistry and Technologies of Drug, Sapienza University, P. le Aldo Moro 5, 00185 Rome, Italy; stefania.garzoli@uniroma1.it; 7Laboratory of Biotechnology and Plant Improvement, Centre of Biotechnology of Sfax, B.P “1177”, Sfax 3018, Tunisia; raniabensaad@gmail.com (R.B.S.); benhsounanis@gmail.com (A.B.H.); 8Department of Environmental Sciences and Nutrition, Higher Institute of Applied Sciences and Technology of Mahdia, University of Monastir, Monastir 5000, Tunisia; 9Zhejiang Provincial Key Laboratory of Chemical and Biological Processing Technology of Farm Products, School of Biological and Chemical Engineering, Zhejiang University of Science and Technology, Hangzhou 310023, China; banzhaojun@zust.edu.cn

**Keywords:** chemical analysis, antibacterial activity, anti-*Candida* and *Salmonella* effect, antibiofilm activity, insecticidal activity, microorganisms, sous vide, carrot

## Abstract

New goals for industry and science have led to increased awareness of food safety and healthier living in the modern era. Here, one of the challenges in food quality assurance is the presence of pathogenic microorganisms. As planktonic cells can form biofilms and go into a sessile state, microorganisms are now more resistant to broad-spectrum antibiotics. Due to their proven antibacterial properties, essential oils represent a potential option to prevent food spoilage in the search for effective natural preservatives. In this study, the chemical profile of *Citrus limon* essential oil (CLEO) was evaluated. GC-MS analysis revealed that limonene (60.7%), *β*-pinene (12.6%), and *γ*-terpinene (10.3%) are common constituents of CLEO, which prompted further research on antibacterial and antibiofilm properties. Minimum inhibitory concentration (MIC) values showed that CLEO generally exhibits acceptable antibacterial properties. In addition, in situ antimicrobial research revealed that vapour-phase CLEO can arrest the growth of *Candida* and *Y. enterocolitica* species on specific food models, indicating the potential of CLEO as a preservative. The antibiofilm properties of CLEO were evaluated by MIC assays, crystal violet assays, and MALDI-TOF MS analysis against *S. enterica* biofilm. The results of the MIC and crystal violet assays showed that CLEO has strong antibiofilm activity. In addition, the data obtained by MALDI-TOF MS investigation showed that CLEO altered the protein profiles of the bacteria studied on glass and stainless-steel surfaces. Our study also found a positive antimicrobial effect of CLEO against *S. enterica*. The anti-*Salmonella* activity of CLEO in vacuum-packed sous vide carrot samples was slightly stronger than in controls. These results highlight the advantages of the antibacterial and antibiofilm properties of CLEO, suggesting potential applications in food preservation.

## 1. Introduction

An integral taxonomic element within the Rutaceae family is the genus *Citrus*, holding significant botanical importance. In vernacular terms, the produce stemming from this genus is commonly identified as ‘citrus’ or citrus fruits. Renowned for their advantageous nutritional, medicinal, and aesthetic attributes, citrus fruits have gained widespread recognition. Evergreen trees, shrubs, and plants (growing up to 15 m in height) are members of the genus *Citrus*. Their leaves are elliptic or ovate and have a leathery texture. Some are pointed. Individual flowers develop in the axils of the leaves. Each flower has five petals, which may be purple or white. The produce takes the form of a hesperidium, characterized as a berry. The genus *Citrus* includes plants that occur naturally in warm temperate climates, especially in the Mediterranean region. Typically, they tend to be vulnerable to frost [1].

The chemical composition of the *C. limon* fruit is common knowledge. Analyses have been conducted for the entire fruit as well as for distinct components such as the pericarp, juice, pomace, and essential oil. It is also known what the leaves of *C. limon* and the fatty oil obtained from its seeds consist of. Several research centres have undertaken the endeavour of scrutinizing the chemical composition of raw materials derived from numerous species, varieties, and hybrids of *C. limon* [2].

When evaluating the biological effects of *C. limon* fruit and juice, the key group of active substances consists of flavonoids. This category includes flavonones such as eriodictyol, hesperidin, hesperetin, and naringin; as flavones like apigenin and diosmin; and flavonols such as quercetin, along with their various forms. Additionally, various other flavonoids, such as orientin and vitexin (flavones), as well as lymphocitrin and spinacetin (flavonols), are distributed throughout the fruit. Noteworthy among the exclusive flavonoids identified in *C. limon* fruit are naringin, hesperidin, and neohesperidin. Furthermore, *C. limon* distinguishes itself by exhibiting the highest concentration of eriocitrin in comparison to other *Citrus* species [3]. Monoterpenoids stand out as the primary components constituting the essential oil of *C. limon*. The essential oil extracted from the pericarp showcases a predominant quantitative presence of specific compounds, including limonene, β-pinene, γ-terpinene, sabinene, myrcene, geranial (E-citral), neral (Z-citral), and linalool. Apart from terpenoids, the essential oil includes linear furanocoumarin, specifically psoralens, and polymethoxylated flavones [4,5,6].

Despite containing mainly monoterpene components, *Citrus limon* essential oil (CLEO) has a low antioxidant capacity [7]. CLEO, however, exhibits biological action against several types of bacteria. Randazzo et al. [8] demonstrated that *Listeria monocytogenes* is inhibited by oxygenated CLEO monoterpenes. Other species investigated included *Salmonella* species, *Pseudomonas aeruginosa*, and *Staphylococcus aureus* [9,10]. According to Høiby et al. [11], a bacterial biofilm is an organized group of microorganisms immersed in their own matrix consisting of polysaccharides, proteins, and DNA. It is characterized by increased resistance to antimicrobial drugs. According to studies, microbial cells in biofilms are 10 to 1000 times more resistant to antibiotics than planktonic cells [12]. Moreover, 80% of persistent and recurrent human microbial illnesses are caused by bacterial biofilms [13].

Numerous investigations demonstrated that diverse plant products and bioactive constituents from plants possess the capacity to impede the growth and formation of bacterial biofilms while also exhibiting the capability of disintegrating pre-existing biofilms [14,15,16,17]. For food safety, packaging, and preservation, CEOs are affordable, environmentally beneficial, and natural substitutes for synthetic preservatives. CEOs have proven antibacterial properties that can be used in food packaging and preservation. CEOs have the ability to utilize by-products of *Citrus* species in the food industry, replace synthetic antimicrobials, and reduce environmental pollution. In addition, in the agricultural sector, the antibacterial properties of CEOs can be useful in controlling postharvest diseases of fruits and vegetables [18].

Fresh fruit and vegetables have a short shelf life. Fresh produce deteriorates by approximately 50% during harvesting, handling, storage, and transport. EOs are applied to whole and freshly cut fruits and vegetables and are essential in preventing this condition [19]. Lemon EO for use in vegetable products was developed by Sessa et al. [20]. The use of CLEO increases the antibacterial activity of vegetable products and extends their shelf life. Other authors also showed positive results in edible coatings. When lemon EO was applied to strawberries, Perdones et al. [21] showed that the quality of the fruit was maintained throughout storage. Due to their ability to preserve food and their ability to provide maximum protection without affecting the sensory and organoleptic properties of the food, EOs were recently proposed for use in food-related applications. Currently, EOs contain fresh fruits and vegetables that are hardly processed [22,23].

Many investigations highlighted the insecticidal capabilities of CLEO extracted from various citrus species and their components, observed at various intervals. Some of these essential oils are commercially available and employed by consumers for insect pest management [24]. As a result of this problem, the search for a safe, plant-based solution was born [25]. Since citrus plants have medicinal properties, they could replace traditional chemical pesticides in this situation. Therefore, bioactive compounds of citrus, especially CEO, can be used as alternatives to insecticides created from synthetic and inorganic chemicals [26,27].

Understanding the effect of cooking products on the development of microorganisms is crucial for the assessment of microbiological safety [28]. Numerous studies show that microorganisms found in foods prepared by the sous vide method during consumption are the result of the raw materials surviving the cooking process [29]. Most pathogenic bacteria grow best at temperatures between 30 and 50 °C, which also marks the beginning of the suppression of bacterial growth and reproduction [4,5,6]. Therefore, to ensure that food pathogens, including *Salmonella* species, *L. monocytogenes*, and pathogenic strains of *E. coli*, are inactivated, food should not be processed at a core temperature lower than 54.4 °C and cooking should take no longer than 6 h [30]. However, this point is only relevant if the food was not pasteurized before.

Our study is unique in that it employed the EO *C. limon* to determine its antibacterial efficacy against *S. enterica* on a food model, specifically carrots. The inoculation of mostly meat was the focus of preliminary investigations. The first objective of our study was to evaluate the chemical composition of CLEO from fresh petals and its antimicrobial (in vitro, in situ), antibiofilm and insecticidal activities. In the second test, we investigated the microbiological analysis of the sous vide treatment of the vegetable with a combination of *C. limon* and inoculation with *S. enterica*.

## 2. Results and Discussion

### 2.1. Chemical Composition of Citrus limon Essential Oil (CLEO)

The findings from the analysis of the chemical composition of the CLEO sample are documented in Table 1.

Forty-five volatile constituents representing 99.7% of total essential oil were identified (Table 1). The most abundant constituents are limonene (60.7%), *β*-pinene (12.6%), and *γ*-terpinene (10.3%). On the other hand, *α*-pinene (2.6%), sabinene (2.4%), and geranial (2.2%) were identified in minor quantities. Other volatile compounds are presented in amounts less than 2.0%.

Overall, EO obtained by cold pressing the fresh pericarp of green lemons is characterized by a high percentage of monoterpene hydrocarbons (13 constituents, 92.4% of the total) (Table 1). On the other hand, the total percentage amount of sixteen oxygenated monoterpenes observed was 5.4%, considering that four monoterpene aldehydes contributed an amount of 3.4%. The results presented in Table 1 indicate that the sesquiterpenes identified were only sesquiterpene hydrocarbons (1.8% of the total).

The data obtained from the qualitative–quantitative analyses demonstrate that the sample of CLEO is a complex mixture of terpene and non-terpene volatile constituents. It is important to point out that limonene, as a monoterpene hydrocarbon, is the most abundant compound. Additionally, the CLEO sample is a limonene chemotype of essential oil. The results of previously published studies conducted on CLEO also indicate a relatively high content of monoterpene hydrocarbons [31,32,33,34,35]. In addition, it should be noted that among all previously examined essential oils, limonene was identified as the highest concentration. Himed et al. [34] reported that the CLEO of the Eureka variety contains 67.1% limonene, while Espina et al. [36] showed that lemon essential oil of the Fino or Verna variety possesses a slightly lower concentration of limonene (59.1%). It must be pointed out that the chemical composition of the essential oil can vary depending on the method of extraction, variety, as well as maturity stage of the used plant material [32,34,36]. A previous report indicated that immature fruit represents a limonene chemotype [32]. On the other hand, at a semimature stage, the limonene level decreased while the *β*-pinene level increased up to 31.5% of the total, which corresponds to the limonene *β*-pinene chemotype.

### 2.2. Antimicrobial Activity of CLEO In Vitro

Several studies consistently demonstrated the inhibitory effects of *Citrus*, with a significant portion of the research focusing on its antibacterial activity [37]. In this study, the potential of CLEO to impede the growth of various bacterial and fungal strains was assessed through disc diffusion and MIC methods. G^+^ bacteria exhibited varying degrees of resistance, with *M. luteus* being the most resilient and *B. cereus* the most susceptible. Among G^−^ bacteria, *V. parahaemolyticus* displayed the highest resistance, while *E. coli* and *Y. enterocolitica* were the most sensitive. *C. krusei* exhibited the most notable antimicrobial efficacy against yeast compared to other cases studied (refer to Table 2). In a separate study, *C. limon* EO demonstrated notable effects on *Staphylococcus aureus*, *Klebsiella pneumoniae*, *Enterococcus faecalis*, and *Salmonella*, Paratyphi A. [38]. In the study by Ben Hsouna et al. [31], compared to the tested strain, CLEO showed variable levels of antibacterial activity. The inhibitory zones observed in this study ranged from 13 to 26 mm. Notably, the largest inhibitory zone was recorded against *L. monocytogenes* (26 mm) within G^+^ bacteria, followed by *B. cereus* (24 mm) and *S. aureus* (22 mm). Among G^−^ bacteria, the most robust inhibitory zone (18 mm) was identified against *S. enteritidis*, although the results indicated comparatively smaller inhibition zones compared to our study. In the study provided by Kehal et al. [39], the antimicrobial activity evaluated by measuring the diameter at which the growth of microorganisms is inhibited reveals that *P. aeruginosa* and *E. coli* bacterial strains are resistant to lemon essential oil, with *E. coli* displaying a slight sensitivity and an inhibition zone diameter of 8.35 mm. However, according to Hayes and Markovic [40], CLEO has antibacterial properties against *E. coli, S. aureus, P. aeruginosa*, and *C. albicans*. Conversely, Espina et al. [36] demonstrated that orange and lemon oils had no effect on the six examined microorganisms, but mandarin oil moderately inhibited the growth of G^+^ bacteria, which is in line with our findings regarding lemon oil.

Results obtained from the disc diffusion method prompted a further antimicrobial assessment of CLEO. With that in mind, the micro-dilution assay was performed, and the obtained results are presented in Table 3. Through the study of the minimum inhibitory concentrations, MIC_50_ and MIC_90_ values were found. Out of tested G^+^ bacterial strains, *M. luteus* was found to have low MIC_50_ (2.33 mg/mL) and MIC 90 (2.52 mg/mL) values. For *S. aureus*, the highest MIC_50_ and MIC_90_ values were found. The most sensitive strain among G^−^ bacteria was *V. parahaemolyticus* with MIC_50_ of 6.23 mg/mL and MIC_90_ of 6.35 mg/mL, while the most resistant was *E. coli* with MIC 50 of 22.61 mg/mL and MIC 90 of 22.80 mg/mL. The lowest minimal inhibition concentration from yeasts was found against *C. glabrata* (MIC_50_ of 3.33 mg/mL and MIC_90_ of 3.48 mg/mL) and highest against *C. albicans* (MIC_50_ of 23.33 mg/mL and MIC_90_ of 24.03 mg/mL). Our findings indicate that G^+^ bacteria are more sensitive to the examined oil, with a MIC_50_ range of 2.33 to 6.19 mg/mL, than G^−^ bacteria, which showed a range of 6.23 to 22.61 mg/mL. Interestingly, the oil demonstrated antibacterial action in the concentration range in which other essential oils considered among the most active showed their antimicrobial effect [41,42]. The literature data show that the CLEO obtained from the leaf, at lower concentrations (0.025–0.1 mg/mL), exhibited no antibacterial effect against six tested microorganisms. However, when applied at higher concentrations, the essential oil significantly inhibited *E. faecium, B. cereus, S. aureus*, and *E. coli* but again had no action against *S. typhi* and *S. dysentery* [43]. Different studies demonstrated the ability of citrus essential oils to suppress microorganisms [18]. Yi et al. [44] reported that CLEO has an inhibitory effect against *E. coli*. According to Deng et al. [45], a ranking of the sensitivity of certain bacteria to *C. paradisi* EO is as follows: *P. aeruginosa > S. aureus > S. typhimurium > B. subtilis > E. coli*. Our findings found that EO had a stronger antibacterial impact against *S. enterica*. Additionally, our study reveals that the most sensitive to the tested CLEO was *V. parahaemolyticus*. Citrus EOs may have shown strain-dependent antibacterial action, in agreement with what was reported by Yang et al. [46]. Orange, lemon, mandarin, and grapefruit EOs were reported to have reduced inhibitory action against *L. curvatus* and *L. sakei* by Viuda-Martos et al. [47]. According to Ambrosio et al. [48], citrus terpenes and the EO phase of orange exhibited minimal inhibitory activity against beneficial bacteria like *Lactobacillus plantarum* and *L. rhamnosus*. In contrast, these compounds displayed heightened inhibitory effects against pathogenic bacteria, including *E. coli*, *S. aureus*, *E. faecalis*, and *L. monocytogenes*. Conversely, these compounds demonstrated the highest inhibitory activity against pathogenic bacteria, including *E. coli*, *S. aureus*, *E. faecalis*, and *L. monocytogenes*.

### 2.3. In Situ Antimicrobial Activity of CLEO in Vapour Phase

Bacterial species can significantly reduce plant quality and quantity in grains, pulses, fruits, and vegetables during production, transit, and storage, resulting in an annual crop loss of 20–40% [49]. In order to protect food from contamination by foodborne pathogenic bacteria, foods must have a longer shelf life. The food contamination caused by outdated food preservative methods cannot be controlled; therefore, new approaches, such as solid and vapour phase techniques, are needed to preserve the food products’ shelf life [50]. Furthermore, concerns about the release of harmful compounds induced by the antagonistic effects of chemical preservatives in packaged food products prompted efforts to reduce their usage in the food industry. Given the robust antibacterial properties of EOs, they emerge as promising candidates for natural food preservation. EOs have undergone comprehensive testing through various methods to assess their potential, with many of these tests involving direct contact between EOs and the test organisms. This approach allows for a direct characterization of the inhibitory actions of EOs [50,51,52,53,54]. Nevertheless, the objective of this study was to investigate the antibacterial effects of the tested EO in the vapour phase, representing a subsequent stage in the research due to its promising antibacterial properties. The impact of CLEO was assessed against yeasts, G^−^, G^+^ bacteria, and biofilm-forming G^−^ bacteria that proliferate on carrots, kohlrabi, and apples (refer to Table 4 and Figure 1).

In an apple model contaminated with G^+^ bacteria, 500 µg/mL of CLEO was the most efficient against *S. aureus* (65.41%), while 62.5 µg/mL and 125 µg/mL applied quantities of CLEO demonstrated pro-bacterial effects in the suppression of *M. luteus*. In the assessment of G^−^ bacterial strains, the vapour phase of CLEO exhibited optimal efficacy at 500 µg/mL, inhibiting the growth of *Y. enterocolitica* by 74.63% and *V. parahaemolyticus* by 57.32% at 62.5 µg/mL. However, an intriguing observation was made at 250 µg/mL, where a pro-bacterial activity of CLEO was noted against *Y. enterocolitica*, resulting in a growth stimulation of 56.40%. In the apple model, CLEO demonstrated its highest efficiency against *C. glabrata* and *C. krusei* at a concentration of 62.5 µg/mL, with inhibition rates of 95.54% and 94.81%, respectively. Additionally, *S. enterica*, a biofilm-forming bacterium, experienced the most substantial inhibition at a CLEO concentration of 62.5 µg/mL, amounting to 55.39%. Analysing the inhibitory effects on G^+^ bacterial strains in the carrot model, it was observed that CLEO was most effective against *S. aureus* (75.90%) at a concentration of 500 µg/mL, while *B. cereus* (74.04%) and *E. faecalis* (35.50%) exhibited maximum suppression at concentrations of 62.5 and 125 µg/mL, respectively. Notably, at the highest dosage (500 µg/mL), the vapour phase of CLEO proved most effective against G^−^ bacteria, with reported inhibitory effects of 93.34% and 75.02% against *Y. enterocolitica* and *E. coli*, respectively, in the carrot model (refer to Figure 1). At an applied dosage of 62.5 µg/mL, CLEO exhibited the most potent antibacterial activity against *C. tropicalis* (94.41%), *C. glabrata* (95.54%), and biofilm-forming bacteria *S. enterica* (65.98%).

While *M. luteus* and *B. cereus* were only suppressed when the lowest concentration of the oil was employed, the strongest effect of the vapour phase of CLEO against G^+^ bacteria growing on kohlrabi was obtained in in situ study at the tested concentration of 500 µg/mL against *S. aureus* (85.53%) (Table 4). At 500 µg/mL, noted was the greatest inhibition of *Y. enterocolitica* (93.48%) and *V. parahaemolyticus* (84.19%) from the G^−^ species. The lowest concentration of CLEO produced the strongest inhibition of *C. albicans* (76.63%) and *C. krusei* (97.56%) (Figure 1). At a dosage of 62.5 µg/mL, the vapour phase of CLEO was found to have the highest inhibitory impact on biofilm-forming *S. enterica* in an in situ study using the food model kohlrabi. Phytochemical components are thought to be linked to the antibacterial activity of EOs. The flavouring and preservation industries use EOs extensively because they are fragrant and volatile components of plants’ secondary metabolism [55]. The main constituents of essential oils, monoterpene or sesquiterpene hydrocarbons and their oxygenated derivatives, were recently shown to possess potential antibacterial properties by several researchers [56].

Consistent with the findings of the current study, previously published research consistently highlighted the preservative efficacy of EOs. Significantly, there are reports indicating that the application of cinnamon oil at a concentration of 0.3% extended the storage life of bananas by as much as 28 days, simultaneously reducing the incidence of fungal diseases in the bananas [57]. *Thymus capitata* (0.1%) and *Citrus aurantifolia* (0.5%) oil were found to decrease disease incidence in papaya fruit [58]. Bhuchanania’s shelf life was increased through seed treatment and fumigation with *Ocimum cannum* oil (1 μL/mL) [59]. The use of *Clausena pentaphylla* and *Chenopodium ambrosioides* oils as fumigants in glass containers and natural fabric bags proved effective in safeguarding pigeon pea seeds from infection by *Aspergillus flavus*, *A. niger*, *A. ochraceus*, and *A. terreus* for a duration of up to six months [60,61]. Pigeon pea seeds could also be stored for up to six months using powder-based formulations of *C. pentaphylla* and *C. ambrosioides* oils [62]. Table grapes’ shelf life was extended by up to nine days when *Artemisia nilagirica* oil was used as a fumigant in cardboard [63]. In a similar vein, *Lippia alba* oil increased the shelf life of *Vigna radiata* by up to six months and prevented fungal growth and aflatoxin generation when applied as an air dosage treatment in glass containers [64].

### 2.4. Antibiofilm Activity of CLEO against Biofilm Forming S. enterica

Bacteria undergo a process known as biofilm formation, involving the aggregation of microbial communities characterized by extracellular polymeric molecules. This intricate process enables bacterial cells to establish robust bonds and firmly adhere to various surfaces, both living and non-living. Biofilm formation provides bacteria with a protective environment, allowing them to withstand challenging conditions, including host defences and exposure to antibiotics [65]. The pathogen, either individually or as a community, can endure and develop increased resistance to antibiotic drugs as a result of biofilm formation. The method with crystal violet showed that CLEO induces antibiofilm activity towards *S. enterica* at MIC_50_ of 1.37 and MIC_90_ of 1.47 mg/mL (Table 3). Using mass spectrometry, a MALDI-TOF MS Biotyper was used to evaluate the antibiofilm effect of CLEO on stainless steel and glass surface against *S. enterica*. The spectra derived from planktonic cells and untreated biofilm, serving as control samples, exhibited identical development. Consequently, the spectra originating from planktonic cells were used as representative spectra for the control group. Comparative analyses were conducted with experimental groups subjected to CLEO supplementation on both stainless steel and glass surfaces. Notably, discernible differences between the experimental groups and the control planktonic spectrum emerged early in the experiment (on the 3rd day). These differences were manifested in terms of both the number of peaks and the shape of the mass spectrum, with the spectrum from the stainless-steel surface exhibiting a reduced number of peaks compared to the experimental spectrum from the glass surface (Figure 2A). As the experiment progressed to day 5 (Figure 2B), distinctions persisted in comparison to the control planktonic spectrum, revealing similar shapes and numbers of peaks. This pattern of differences endured throughout the entire experimental duration (Figure 2C–F), except on day 9, when the spectra exhibited considerable similarity. The observed disparities between the experimental groups and the control spectra suggest the degradation and inhibition of the *S. enterica* biofilm resulting from the CLEO application. These findings indicate that CLEO has the potential to disrupt biofilm homeostasis during its initial stages, providing a promising alternative for combating the formation of *S. enterica* biofilm.

Early on in the biofilm growth process, citrus essential oils (CEO) are very active when planktonic cells start to lose their potency and sessile microcolonies start to form [66]. Research also demonstrated that citrus essential oils made from the peel of lemons are capable of eliminating the biofilm that *Streptococcus mutans* form [67]. CEOs extracted from grape grapes can similarly prevent *Pseudomonas aeruginosa* from using its QS system [68]. Additionally, it was shown that EOs derived from *Citrus limonum* and *aurantium* demonstrated their effectiveness in combating multispecies biofilms [69]. Research demonstrated that CEOs derived from grapes can break down the biofilm that *P. aeruginosa* forms by downregulating the quorum-sensing (QS) mechanism [68]. Additionally, lemon oils have the capacity to inhibit pathogenic bacteria, such as the *P. aeruginosas* quorum-sensing mechanism [70].

Some of the previous findings regarding the antibiofilm activity of *Citrus* species show that in the case of *C. reticulata,* its main component, limonene, was a key factor in preventing the formation of the biofilm. After the biofilm was treated with the essential oil, it was shown that it was able to significantly change the morphological features and cell collapse of the biofilm. In a study examining limonene’s ability to suppress biofilm, it was shown that this monoterpene is effective in inhibiting *C. albicans* biofilm to the extent of 87% [71]. Additionally, in the work provided by Gupta et al. [72], it was stated that limonene is an effective antibiofilm agent against *B. cereus*, *E. coli*, and *P. anomala*. In the same study, authors provided information that limonene can eradicate *S. pyogenes, S. mutans*, *S. mitis*, and different strains of *S. aureus* up to an extent that varies in the range of 75–95% [72].

*C. limon* was able to inhibit 100% of *S. epidermidis* biofilms with its essential oil pomelo flavedo (coloured layer or zest of a pomelo fruit) at a concentration of 15.63 µg/mL [73]. It was supposed that limonene from *C. limon* could have antibiofilm effects on *S. epidermidis*. The result is not exclusive to limonene because the EO has a complex chemical composition and may contain additional chemicals. Limonene, as the most abundant constituent of the essential oil of *C. limon*, was responsible for inhibiting the formation of biofilms [73]. In accordance with a surface-coating experiment, limonene decreases bacterial attachment to surfaces and obstructs the subsequent pathways for biofilm growth [72]. However, it was discovered that limonene from CLEO had only a modest effect (MBEC = 15.63–62.50 μg/mL) on the growth of the *S. epidermis* biofilm. Analysis revealed that hydrophobic compounds could pass through the *S. epidermis* G^+^ peptidoglycan cell wall and into the membrane. Biofilm inhibition is made possible by limonene and the essential oil’s hydrophobicity. It is accomplished by breaking down the cell wall and upsetting the respiratory chain, which causes vital cell components to flow out [74].

Using MSP distances, a dendrogram was created to depict the similarities in biofilm structure between the control and experimental groups (Figure 3). Remarkably, the shortest MSP distances were observed during the early stage of the experimental biofilm groups on the 3rd day (3 SECLG), aligning with the control group (3 PCSE). With the increased exposure time of the experimental groups to CLEO, their MSP distances also increased. The most significant difference was noted on the 9th day of the experiment, where the MSP distance of the experimental group was maximal, particularly for the experimental group on the glass surface (9 SECLG). Based on these observations, it is inferred that CLEO has an impact on the homeostasis of the *S. enterica* biofilm, contributing to its inhibition. These findings align with the outcomes of the mass spectra analysis. In the study of Kačániová et al. [75], the antibiofilm activity molecular profile of *C. aurantium* was evaluated. Using dendrograms obtained by MSP spectra to show the grouping patterns of *S. maltophilia* and *B. subtilis*, the genetic similarity was investigated in relation to the molecular differences of biofilm development on various days. Early growth variants of *S. maltophilia* exhibited differentiated branches in both planktonic cells and all experimental groups. In contrast, when comparing the grouping pattern of *B. subtilis* to the media matrix, a time span preference can be observed, with no discernible variation between variants.

### 2.5. Insecticidal Activity of CLEO

Various dangerous insects can be effectively controlled using conventional chemical pesticides. Nonetheless, persistent and unchecked pesticide usage might cause these insects to adapt, which can result in pesticide resistance [76]. Chemical-based pesticides always affect the ecosystem in a perishable way, endangering not only non-target creatures but also the entire food chain [77,78]. As a result of this problem, the search for a different, safe, plant-based solution was born [25]. Since citrus plants have therapeutic qualities, they could take the place of traditional chemical pesticides in this situation. Consequently, bioactive components of citrus, particularly CLEOs, can be employed as an alternative to insecticides generated from synthetic and inorganic chemicals [26,79].

Table 5 presents the results of the evaluated insecticidal activity of CLEO against *H. axyridis*. The results show that at applied doses of 100% and 50%, the tested EO has the greatest insecticidal effect. However, CLEO did not demonstrate potent repellent qualities when applied to *H. axyridis* at concentrations of 6.25% and 3.125%. CLEO exhibited a 25% effect on 10% of the *H. axyridis* population, with 12.5% of the CLEO showing activity on 50% of the insects.

Multiple studies highlighted the outstanding insecticidal effectiveness of CLEO. *Citrus* essential oil (CEO) bioactive chemicals are highly valued for their innovative applications as herbicides, antibacterial agents, and pest control medications due to their well-known antibiofilm action [80]. Similar results were obtained when testing the EO of *Citrus sinensis* against *Musa domestica* larvae and pupae. In the contact toxicity and fumigation assay against housefly pupae, the oil’s percentage inhibition varied between 27.3% and 72.7% for contact toxicity and from 46.4% to 100% for fumigation [81].

### 2.6. Microbiological Analyses of Carrot in Sous Vide Application with S. enterica and CLEO

Many vegetables were studied using sous vide technology, but fruits were not [82,83,84]. Owing to differences in thermal diffusivity, vegetables must be cooked sous vide at specific temperatures in order to destroy *Salmonella* and *E. coli*, two of the most common foodborne pathogens [85]. Herein, the microbiological analyses of raw carrot were performed. On XLD agar, the count and presence of *S. enterica* were confirmed. The initial total count of bacteria (TCB) was 2.23 ± 0.03 log CFU/g, with zero coliform bacteria detected on day 0. The microbiological quality of vacuum-packed carrots was assessed for total bacterial count on the 1st and 7th days of storage (Figure 4). In the control group on the first day, the total bacterial count ranged from 1.03 (at 60 °C for 20 min) to 2.52 log CFU/g (at 50 °C for 5 min), and on the 7th day, it ranged from 1.39 (65 °C for 5 min) to 2.86 log CFU/g (50 °C for 5 min). In the vacuum-packaged group, TCB ranged from 1.08 (at 55 °C for 20 min) to 2.48 log CFU/g (at 50 °C for 5 min) on the 1st day and from 1.03 (55 °C for 20 min) to 2.42 log CFU/g (50 °C for 5 min) on the 7th day. The vacuum-packaged group treated with CLEO showed TCB ranging from 1.06 (55 °C for 20 min) to 1.94 ± 0.05 log CFU/g (50 °C for 5 min) on the 1st day and from 1.16 (at 50 °C for 20 min) to 1.66 log CFU/g (at 50 °C for 5 min) on the 7th day. The group treated with *S. enterica* had TBC ranging from 1.09 (60 °C for 10 min) to 2.48 log CFU/g (50 °C for 5 min) on the 1st day and from 1.52 (at 50 °C for 20 min) to 2.39 log CFU/g (at 50 °C for 5 min) on the 7th day. In the group treated with CLEO and with the addition of bacteria *S. enterica*, TBC ranged from 1.17 (55 °C for 20 min) to 2.40 log CFU/g (50 °C for 5 min). Notably, the number of bacteria was reduced in the groups treated with CLEO and in the group with CLEO and *S. enterica*.

Several herbal products were identified as carriers of *Salmonella* bacteria. A comprehensive review highlighted an 8% prevalence of *Salmonella* in vegetables [86]. In a study utilizing a tomato model, the efficacy of plant essential oils (PEO) against *Salmonella* was evaluated. The findings of the study indicated that the volatile components of PEO led to a substantial reduction in the microbial load of the target pathogens on tomato samples, reaching a limit of approximately 6.0 log 10 CFU/mL [87]. The observed biological efficacy of plant essential oils in vegetable dishes is likely influenced by both the inherent qualities and external factors affecting the sample products [88]. The biological efficacy of PEOs may be influenced by the low-fat content of vegetables [89]. In the alfalfa seed food model, certain volatiles from PEOs demonstrated the effective inhibition of *Salmonella* species [90]. Myrtle PEO was evaluated for its effectiveness against antibiotic-resistant *Salmonella* species using tomato and lettuce food models. The results indicated a significant reduction in the microbial count when treated with myrtle PEO, highlighting its potential as an antimicrobial agent [91]. A separate study demonstrated that certain plant volatiles, including carvacrol and cinnamon aldehyde, exhibited a substantial reduction in colony-forming unit (CFU) counts of target pathogens in kiwi food models. However, their inhibitory impact was weaker in the food model of honeydew melon. These variations could be attributed to the inherent or extrinsic qualities of the respective foods [92]. Using a lettuce food model, the biological ability of oregano PEO against *Salmonella* species was assessed.

These results validated the significant effectiveness of oregano PEO, indicating its potential to function as a natural substitute for chemical washing solutions in maintaining the quality of food products contaminated with food-borne pathogenic bacteria; Gündüz et al. [93]. Furthermore, the assessment of oregano PEO’s antimicrobial efficacy against resistant *Salmonella* species on a tomato food model verified that the herb was successful in combating the test-resistant *Salmonella* pathogen and significantly decreased the number of counts viable in the tomato food model, with a significance level of 2.78 log CFU/g Gündüz et al. [94].

Microbiological investigations were carried out by Rinaldi et al. [95] on steamed and sous vide carrots as well as Brussels sprouts after refrigeration for 1, 5, and 10 days. The analyses encompassed aerobic and anaerobic total plate counts, mesophilic lactic acid bacteria, yeasts, and microscopic fungi. Remarkably, even after 10 days of storage at 4 °C, the microbiological counts for both groups of carrots remained consistently below 1 log colony-forming unit per gram (CFU/g). As a result, it seemed that both heat treatments effectively reduced the initial counts. Similarly, carrots that were prepared sous vide for up to 30 days in chilled storage showed aerobic total plate counts of less than 1 log CFU/g, according to Sebastiá et al. [96]. Regarding Brussels sprouts, samples prepared via both sous vide and steam cooking showed a reduction in all microbial counts. For steamed samples, the aerobic total plate counts were 3.46 log CFU/g, but for sous vide sprouts, the values ranged from 2.34 to 3.15 log CFU/g. Furthermore, after being refrigerated for up to 10 days, sous vide Brussels sprouts showed all other microbial counts (< 1 log CFU/g) lower than those seen for steamed samples. The authors also linked these outcomes to the correct timing and temperature of the steps, in particular, the regulated heating and cooling phases that eliminated the initial flora more effectively than steaming [96]. In our study, lower temperatures were used as in previous studies. From our results, we found that TBCs were lower than 1 log CFU/g at a temperature of 65 °C.

The number of coliform bacteria (CB) on day 1 in the control group, vacuum-packed control group, and CLEO group was not identified (Figure 5). On day 7, in the control samples, CB ranged from 1.19 (50 °C for 10 min) to 1.42 log CFU/g (50 °C for 5 min); the vacuum packaged and in the group with coliform bacteria CLEO was not identi-fied. In the group with the addition of *S. enterica* on the first day, CB ranged from 1.88 (55 °C for 5 min) to 2.53 log CFU/g (50 °C for 5 min), and on the seventh day, ranged from 1.62 (50 °C 20 min) to 1.96 log CFU/g (50 °C for 5 min). In the group treated with CLEO and with *S. enterica*, CB ranged from 1.74 (50 °C for 20 min) to 2.35 log CFU/g (50 °C 5 min) on the first day and on the seventh day from 1.61 (50 °C for 10 min) to 1.79 log CFU/g (50 °C 5 min).

Vegetables are frequently contaminated with *S. typhi, E. coli, Enterobacter* spp., *Klebsiella* spp., *Serratia* spp., *Providencia* spp., *P. aeruginosa, S. aureus*, as well as other potentially dangerous bacteria [97]. Apart from these, certain other varieties of vegetables are more vulnerable to deterioration due to various microbes such as *Salmonella* spp., *B. cereus, C. jejuni, C. botulinum, E. coli* O157:H7, *L. monocytogenes*, and *V. cholera*. Most of them are facultative anaerobes, meaning that the cells can grow and thrive in both oxygen-filled and non-oxygen environments [98].

In carrot sous vide samples across all groups on the first day, a total of 245 species were isolated. Among all isolates, 11 families, 15 genera, and 28 species were identified. The predominant species were *S. enterica* (14%) and *R. radiobacter* (9%), with *Stenotrophomonas* spp. (7%) and *P. brassicacearum* (6%) also being prominently identified across all groups (Figure 6).

In the carrot sous vide samples across all groups over seven days, a comprehensive isolation effort led to the identification of a total of 268 species. Among these isolates, 10 families, 14 genera, and 24 species were recognized. The predominant species included *S. enterica* (18%) and *R. radiobacter* (10%), trailed by *S. maltophilia* (9%) and *A. calcoaceticus* (6%) across all groups (Figure 7).

The safety of sous vide cooking at temperatures below 55 °C remains uncertain, as there is currently insufficient scientific evidence to support the development of any predictive models for the inactivation of foodborne pathogens in vegetables at or below 55 °C. An alternative could be to apply non-thermal barriers employing some cutting-edge non-thermal technologies, add bio-preservatives in the form of essential oils, or use time–temperature indicators in the packaging to document a product’s storage history [99,100]. Recent research explored the use of EO, known for its natural antibacterial and antifungal properties, in the processing of fresh-cut potatoes. [101]. The incorporation of EO in the processing of fresh-cut potatoes offers a dual advantage. Firstly, the natural antibacterial properties of EO, attributed to components such as camphor, 1,8-cineole, α-pinene, borneol, and verbenone, allow for the elimination of synthetic preservatives. Secondly, the distinctive aroma of the vegetables can be enhanced by combining EO, thereby contributing to the flavor profile of the final product. The study demonstrated that the use of EO, coupled with vacuum packing and refrigerated storage, effectively reduced the growth of mesophilic bacteria and Enterobacteriaceae in minimally processed potatoes intended for sous vide cooking, even after 11 days of storage [102].

## 3. Materials and Methods

### 3.1. Citrus limon Essential Oil

The essential oil (EO) used in this study was obtained by the extraction method by cold pressing the fresh pericarp of green lemons *Citrus limon* (CLEO), which were obtained from Hanus s.r.o. in Nitra, Slovakia. The essential oil was isolated from the Italian Feminello variety and was diligently preserved in darkness at a temperature of 4 °C for subsequent analysis.

### 3.2. GC and GC/MS Chemical Analysis of CLEO Sample

The chemical profile of *C. limon* essential oil was meticulously analyzed employing a 6890 N gas chromatograph coupled with a quadrupole mass spectrometer 5975 B (Agilent Technologies, Santa Clara, CA, USA). For semiquantitative determination, the percentage composition of each identified compound was assessed using a 6890 N gas chromatograph coupled with an FID detector (Agilent Technologies, Santa Clara, CA, USA). Data acquisition and interpretation of both mass spectra and chromatographic information were conducted through the utilization of the HP Enhanced ChemStation software D.03.00.611. (Agilent Technologies, Santa Clara, CA, USA).

To separate volatile constituents, an HP-5MS capillary column [(5%-phenyl)-methylpolysiloxane; 30 m length; 0.25 mm internal diameter; 0.25 µm film thickness)] was installed in the gas chromatography (GC) oven. An injection volume of 1 µL, containing a 10% solution of essential oil in hexane, was used. Helium 5.0 served as the carrier gas with a flow rate of 1 mL/min. The split/splitless injector, MS source, and MS quadrupole were maintained at temperatures of 280 °C, 230 °C, and 150 °C, respectively. The split ratio was set at 40.8:1. Mass spectra were acquired in the mass scan range of 35–550 amu at an ionization energy of 70 eV. The oven temperature was programmed as follows: 50 °C to 75 °C (increasing rate, 3 °C/min) with a hold time of 4 min, 75 °C to 120 °C (increasing rate, 5 °C/min) with a hold time of 2 min, and 120 °C to 290 °C (increasing rate, 5 °C/min). The total run time for the analysis was 57.33 min.

The identification of individual volatile constituents was executed through a dual approach. First, mass spectra were compared with the reference spectra stored in the MS library (Wiley7Nist). Additionally, the retention indices (RI) of the identified compounds were cross-referenced with the retention indices of a series of n-alkanes (C_7_–C_35_) for further confirmation and validation [103]. The percentage values of the components (amounts greater than 0.1%) were derived from the areas of their GC peaks.

### 3.3. Antimicrobial Assay

#### 3.3.1. Tested Microorganisms

The antimicrobial effectiveness of the examined CLEO was assessed against a spectrum of bacterial strains, encompassing Gram-positive (G^+^) *Bacillus cereus* CCM 2010, *Micrococcus luteus* CCM 732, and *Staphylococcus aureus* CCM 3953; Gram-negative (G^−^) bacteria, including *Escherichia coli* CCM 3953, *Vibrio parahaemolyticus* CCM 5937, *Yersinia enterocolitica* CCM 7204T; and yeasts, including *Candida albicans* CCM 8186, *Candida glabrata* CCM 8270, *Candida krusei* CCM 8271, and *Candida tropicalis* CCM 8223. All bacterial strains and yeasts were sourced from the Czech Collection of Microorganisms in Brno, Czech Republic. For antibiofilm activity assessment, biofilm-forming G^−^
*Salmonella enterica* isolated from milk production was used. Both yeast and bacterial inoculum were cultured for 24 h at 25 °C and 37 °C in Sabouraud Dextrose Broth (SDB, Oxoid, Basingstoke, UK) and Mueller Hinton Broth (MHB, Oxoid, Basingstoke, UK), respectively, prior to analysis. The optical density of the yeast and bacterial inoculum was standardized at 0.5 McFarland on the day of experimentation [104].

#### 3.3.2. Disc Diffusion Method

A disk diffusion susceptibility test was performed using the above microbial strains and following the methodology of previous studies [104]. Briefly, Mueller Hinton Agar (MHA; Merck, Darmstadt, Germany), Sabouraud Dextrose Agar (SDA; Merck, Darmstadt, Germany), and density-adjusted bacterial and yeast strains were used for analysis. Blank discs were inoculated with the given microorganism, and inhibition zones were measured after 24 h incubation at a temperature suitable for bacteria and yeast. Gentamicin and cefoxitin antibiotics (ATBs) (Oxoid, Basingstoke, UK) served as controls.

#### 3.3.3. Minimal Inhibition Concentration

MIC_50_ and MIC_90_ values were determined using the previous method on a 96-well microtiter plate [105]. CLEO concentrations ranged from 10 mg/mL to 0.00488 mg/mL in Mueller Hinton broth. After 24 h incubation at 37 °C for bacteria and 25 °C for yeast, absorbance was measured at 570 nm using a Glomax spectrophotometer. The minimum concentration of CLEO inhibiting 50% of bacterial growth (MIC_50_) and the concentration inhibiting 90% of bacterial growth (MIC_90_) were evaluated for each experiment.

### 3.4. In Situ Analyses on the Fruit and Vegetables

The antimicrobial effectiveness of CLEO was evaluated in situ against selected yeast and bacterial strains, including both G^+^ and G^−^ bacteria, using common food items like apple, carrot, and kohlrabi as growth substrates. Following the methodology outlined by Kačániová et al. [106], 0.5 mm slices of apple, carrot, and kohlrabi were dried, cleansed, and placed on agar in 60 mm Petri plates for bacterial inoculation. CLEO samples, dissolved in ethyl acetate at concentrations of 500, 250, 125, and 62.5 mg/L, were applied to sterile filter paper, which was positioned on the Petri dish lid. After one minute of ethyl acetate evapouration, the dishes were sealed and incubated for seven days at 37 °C. Bacterial growth volume density (vv) was determined using the ImageJ program from the National Institutes of Health, Bethesda, Maryland, USA. The volume density of bacterial colonies was determined through the following formula:(1)$$vv(\%)=\frac{P}{p}$$
where P denotes the stereological grid points that strike the colonies, and p denotes the points that fall within the reference space (growth substrate used).

The percentage (%) of bacterial growth inhibition (BGI) resulting from the EOs vapour phase treatment was expressed as follows:(2)$$BGI=\frac{C-T}{C}\times 100$$
where C denotes the control group, and T denotes the treatment group. Bacterial growth expressed as *v/v* is represented by both groups. Findings were obtained because growth stimulation is indicated by negative values.

### 3.5. Biofilm Development Study

#### 3.5.1. Crystal Violet Assay

The determination of Minimal Biofilm Inhibitory Concentration (MBIC) followed the methodology established by Kačániová et al. [75]. Bacterial suspensions were incubated in Mueller Hinton Broth (MHB, Oxoid, Basingstoke, UK) for 24 h at 37 °C under aerobic conditions. Following incubation, an inoculum with an optical density of 0.5 McFarland standard was prepared. A 96-well microtiter plate was then filled with 50 μL of the inoculum and 100 μL of MHB. The first column of the microplate was treated with 100 μL of CLEO. A two-fold dilution, ranging from 100 mg/mL to 0.049 mg/mL, was achieved through pipetting. Maximal growth control was established using MHB with a bacterial inoculum, while MHB with essential oil served as the negative control. After a 24 h incubation period at 37 °C, the supernatant was removed, and the wells underwent three saline washings with 250 μL of water before air-drying for 30 min at room temperature. Post-drying, the wells were stained for 15 min with 200 μL of crystal violet (0.1% *w*/*v*). Following multiple washes with distilled water, the plates were left to dry. To resolubilize the samples, 200 μL of 33% acetic acid was added. The Glomax spectrophotometer (Promega Inc., Madison, WI, USA) was employed to measure the samples at 570 nm. MBIC was defined as the concentration at which the absorbance was equal to or less than the negative control. The concentrations required to prevent 50% and 90% of biofilm development were designated as MBIC_50_ and MBIC_90_, respectively.

#### 3.5.2. Biofilm Formation Detection by MALDI-TOF MS Biotyper

The MALDI-TOF MicroFlex instrument from Bruker Daltonics was used to assess protein degradation during biofilm formation. For this analysis, we followed the procedures and methodology from previous research [106]. The biofilm-forming bacteria *S. enterica* was loaded into polypropylene tubes containing stainless steel and glass microscope slides. CLEO was added at a concentration of 0.1%, with untreated tubes serving as controls. This was followed by incubation at 37 °C for 3, 5, 7, 9, 12, and 14 days with constant agitation. Biofilms collected from steel and glass surfaces were deposited on target plates, and planktonic cells from control samples were examined. MALDI-TOF analysis performed in linear positive mode with a mass-to-charge ratio of 2000 to 20,000 produced 18 standard global spectra (MSP). Based on the Euclidean distances calculated from the generated MSPs, dendrograms were generated for automated analysis.

### 3.6. Insecticidal Activity

*Harmonia axyridis* was utilized as a model organism to assess the insecticidal activity of CLEO. In each group, 100 *H. axyridis* individuals were assigned to the Petri dishes. The lid was sealed with a sterile filter paper circle. By dilution with 0.1% polysorbate, concentrations (100, 50, 25, 12.5, 6.25, and 3.125%) were created. The sterile filter paper was treated with 100 µL with the suitable CLEO concentration. The plates were placed at room temperature for 24 h after being wrapped with parafilm. A total of 100 µL of 0.1% polysorbate was used in the control group. The number of living and deceased population was counted after 24 h. Analyses were performed in triplicate [104].

### 3.7. Sous Vide Vegetable Analyses

#### 3.7.1. Sample Preparation

Carrot samples were used in this investigation. The carrot sample was bought from an authorized store, based on the information on the label made in the Slovak Republic. A total of 2.5 kg of carrots were gathered and chilled prior to being transported to the microbiological laboratory. The carrot samples were then chopped using a sterile knife into 5 g chunks, and each chunk was weighed. A total of 480 five-gram samples were prepared. Three samples of raw carrots were used: 240 samples of treated and control carrots on day 1 and 240 samples of treated and control carrots on day 7. Samples weighing 5 g of chopped carrot were treated with 1% *v*/*w* of CLEO solution, dissolved in rapeseed oil, and vacuum packaged using a Concept, Choce, Czech Republic, vacuum packer. Every carrot sample (5 g) was packaged separately. The control sample groups included both vacuum-packed and unpackaged samples. In order to avoid damaging the carrot sample, the samples containing 100 µL of *S. enterica* and 1% *v*/*w* of CLEO were prepared. The samples were then placed into the main bags and gently mixed for around one minute. Following this procedure, they were vacuum-packed. In the concentration of 1.5 × 10^8^ CFU (0.5 McF), 100 µL of *S. enterica* was added to the sample [107].

The following was made accessible to us during our trial:Control: Fresh carrot sample was treated at 50–65 °C for 5 to 25 min after being packed in polyethylene bags and kept at 4 °C.Control + vacuum: Fresh carrot sample was treated at 50–65 °C for 5 to 25 min after being vacuum-packed in polyethylene bags and kept at 4 °C.EO: vacuum-packed fresh carrot treated with 1% CLEO was kept at 4 °C and treated for 5–25 min at 50–65 °C.*Salmonella*: vacuum-packed fresh carrot treated with *S. enterica* was kept at 4 °C and treated for 5–25 min at 50–65 °C.*Salmonella* + EO: vacuum-packed fresh carrot treated with *S. enterica* and 1% CLEO was kept at 4 °C and treated for 5–25 min at 50–65 °C.

Day zero involved preparing the control samples using raw, uncooked carrot. The samples were macerated for 24 h after the application of EO from the first group of samples, and *S. enterica* from the second group of samples were applied, gently mixed, and combined with the samples. The samples were prepared using the Arnsberg, Germany-based CASO SV1000 sous vide machine. The samples were separated into groups and heat-treated for sous vide preparation under carefully monitored temperature and time parameters. The high-barrier polyethylene vacuum packaging bags were made of 40- to 200-micron waterproof material, which protects against humidity and extreme temperatures (−30 °C to +100 °C). As per the information outlined in the data sheet, these products ensure an exceptionally extended shelf life, making them well-suited for freezer and refrigeration storage over numerous years. They are certified as safe for food storage, characterized by a lack of taste and odor transfer to the stored items. Additionally, the items are explicitly stated to be completely free from plasticizers, including bisphenol A, and are guaranteed to be entirely devoid of microplastics.

#### 3.7.2. Microbiological Analyses

The raw carrot was analysed for microbiological parameters on day 0. Microbiological tests were conducted on days 1 to 7, and samples were stored at 4 °C. Five grams of sample were diluted with 45 mL of sterile saline solution (0.1%) using an Erlenmeyer beaker. The samples were homogenized in Burgwedel, Germany’s GFL 3031 shaking incubator for 30 min. The microbial communities were investigated. Violet Red Bile Lactose Agar (VRBL, Oxoid, Basingstoke, UK) was employed to foster the growth of coliform bacteria, with an incubation period at 37 °C ranging from 24 to 48 h. Plate Count Agar (PCA, Oxoid, Basingstoke, UK) was utilized for cultivating total viable counts (TVCs), and the incubation took place at 30 °C for 48 to 72 h. For the enumeration of *S. enterica*, Xylose Lysine Deoxycholate (XLD, Oxoid, Basingstoke, UK) was inoculated with a 0.1 mL sample, and the incubation duration was 24 h at 37 °C. Subsequently, the media underwent assessment to determine the total viable bacteria, coliforms, and *Salmonella* counts.

#### 3.7.3. Identification of Bacteria

Bacteria from carrot samples were identified using reference libraries and MALDI-TOF MS Biotyper (Bruker, Daltonics, Bremen, Germany). The stock solution of organic material was made of 50% acetonitrile, 47.5% water, and 2.5% trifluoroacetic acid. Sample preparation involved extraction of the biological material, air drying, and treatment with formic acid and acetonitrile before application to a MALDI plate. Mass spectra were generated using a Bruker Daltonics MALDI-TOF Microflex mass spectrometer, and MALDI Biotyper 3.0 software was used for analysis. Identification scores ranging from 2300 to 3000 indicated a highly probable species identification. The methodology and procedures for identification of bacteria were based on a preliminary study [108].

### 3.8. Statistically Evaluation

All assessments were conducted in triplicate, and the results are presented as mean values ± standard deviation (SD). Prism 8.0.1 software (GraphPad Software, San Diego, CA, USA) was employed to perform a one-way analysis of variance (ANOVA), followed by Tukey’s Honestly Significant Difference (HSD) test, with a significance level set at *p* ≤ 0.05. To facilitate analysis, changes in absorbance between measurements were transformed into a set of binary values, utilizing the measured absorbances obtained before and after the experiment. These values were then assigned precise concentrations. A specific formula was devised for this experiment: binary system numbers were designated as 1 (indicating an inhibitory effect) if absorbance values were as low as 0.01, while binary system numbers were set as 0 (representing no effect or a stimulant impact) if absorbance values reached or exceeded 0.01.

The other graphic elaborations were performed by JMP Pro 17.0 software package (SAS Institute, Cary, NC, USA).

## 4. Conclusions

The current study aimed to establish the chemical profile of the essential oil obtained by cold pressing the fresh pericarp of green lemons *C. limon* (CLEO). According to the GC/MS investigation, a significant amount of limonene, *β*-pinene, and *γ*-terpinene was present. More antimicrobial research against pertinent G^+^ and G^−^ bacteria and yeasts was spurred by the presence and quantity of the discovered components, which suggested the possible use of CLEO in the prevention of food spoiling. MIC experiments conducted in vitro demonstrated that CLEO had good antibacterial capabilities in most cases. Furthermore, it was noted that G^−^ bacteria were more vulnerable than G^+^ strains to the CLEO treatment. In situ antimicrobial studies were also conducted to assess the CLEO vapour phase’s capacity to suppress bacterial and yeast growth on particular food models. Here, the growth of *C. glabrata, C. krusei* on apple, *C. glabrata, C. tropicalis* on carrot, *Y. enterocolitica,* and *C. krusei* on kohlrabi were all most successfully prevented by CLEO vapour phase. Compared to the MIC assay for direct-contact application, the G^+^ strains demonstrated a usually greater susceptibility to the CLEO vapour phase compared to G^−^ bacteria. To determine the antibiofilm effectiveness of CLEO against *S. enterica*, biofilm-generating bacteria, additional studies were carried out. Here, the results of the MIC and crystal violet tests showed that CLEO had a strong antibiofilm effect. These results led to mass spectrometry antibiofilm analyses (MALDI-TOF MS), which shed light on the capacity of CLEO to inhibit the growth of biofilms on various surfaces. According to the MALDI-TOF MS study, by disrupting biofilm homeostasis, CLEO significantly altered the protein profile of *S. enterica* on surfaces made of glass and stainless steel. These findings demonstrated that CLEO is a potential antibiofilm agent against this extremely pathogenic bacterium. Overall, the results show how promising CLEO’s antimicrobial and antibiofilm qualities are, providing support for its possible application in food preservation and spoiling management. To ensure the safety and quality of sous vide carrots against *S. enterica*, it is frequently necessary to combine them with other processing procedures, such as using EO and alternative packaging technologies.

## Figures and Tables

**Figure 1 plants-13-00524-f001:**
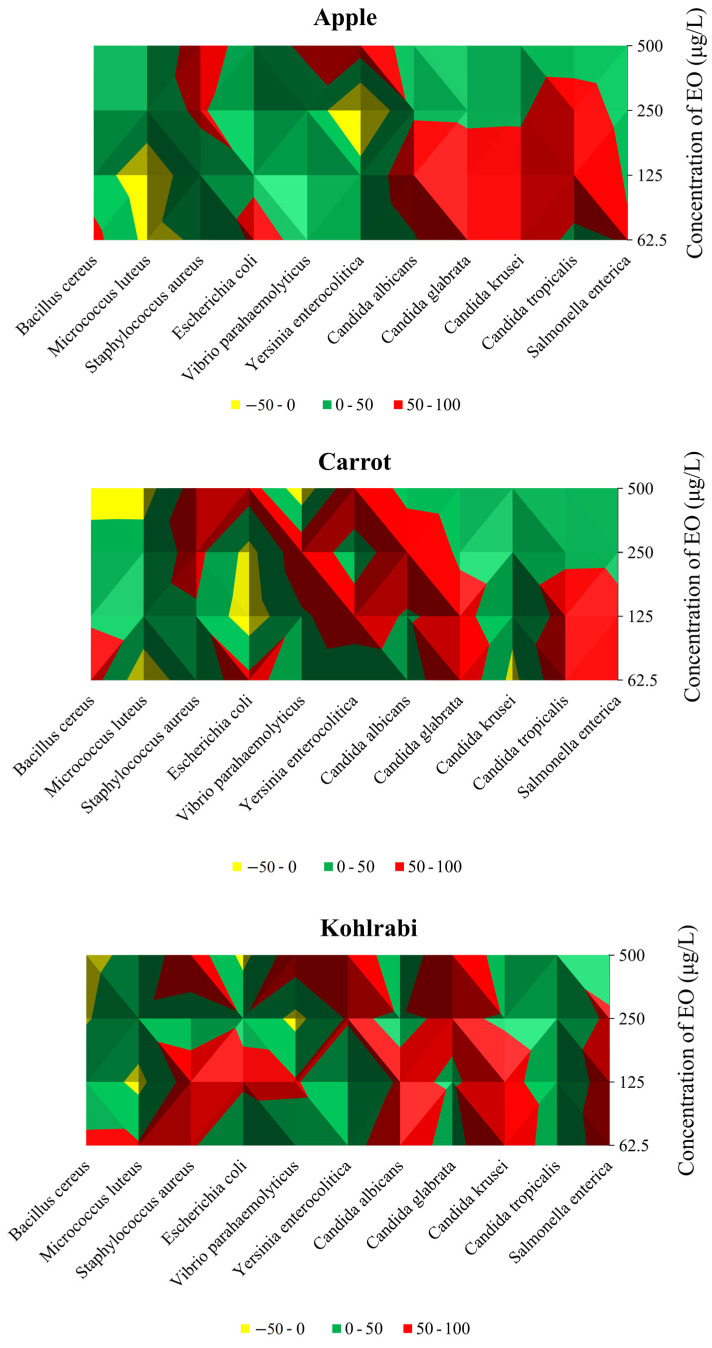
Graphic elaboration of in situ antimicrobial activity (%) in apple, carrot, and kohlrabi.

**Figure 2 plants-13-00524-f002:**
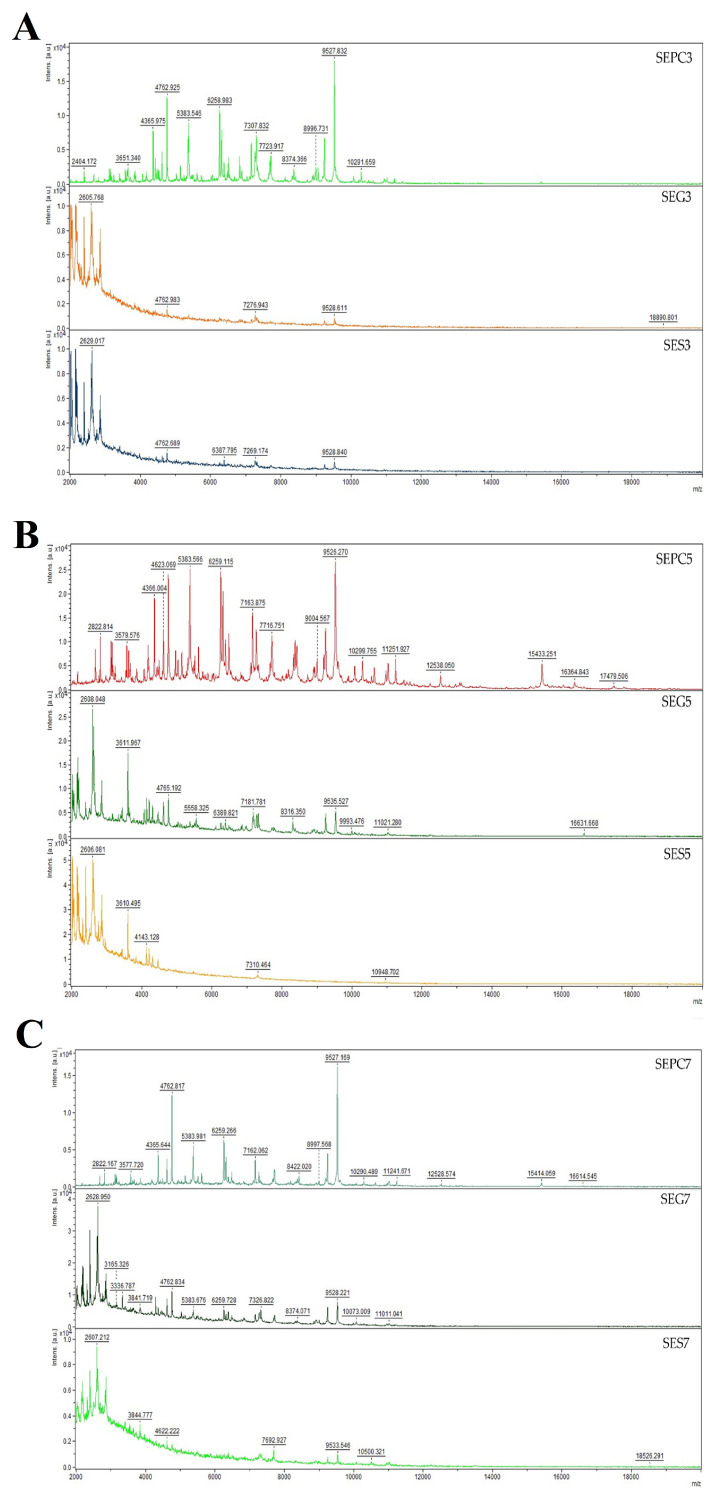
Representative MALDI-TOF mass spectra of *S. enterica*: (**A**) 3rd day, (**B**) 5th day, (**C**) 7th day, (**D**) 9th day, (**E**) 12th day, and (**F**) 14th day. SE = *S. enterica*; G = glass; S = stainless steel; PC = planktonic cells.

**Figure 3 plants-13-00524-f003:**
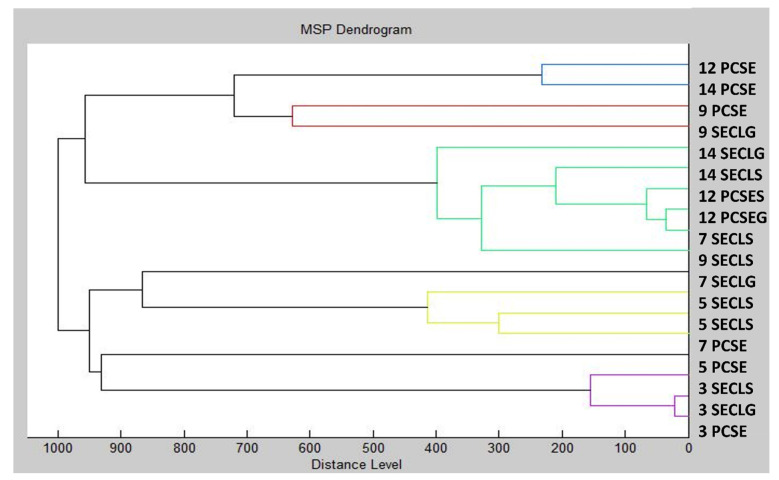
Dendrogram of *S. enterica* generated using MSPs of the planktonic cells and the control. SE = *S. enterica*; G = glass; S = stainless steel; PC = planktonic cells.

**Figure 4 plants-13-00524-f004:**
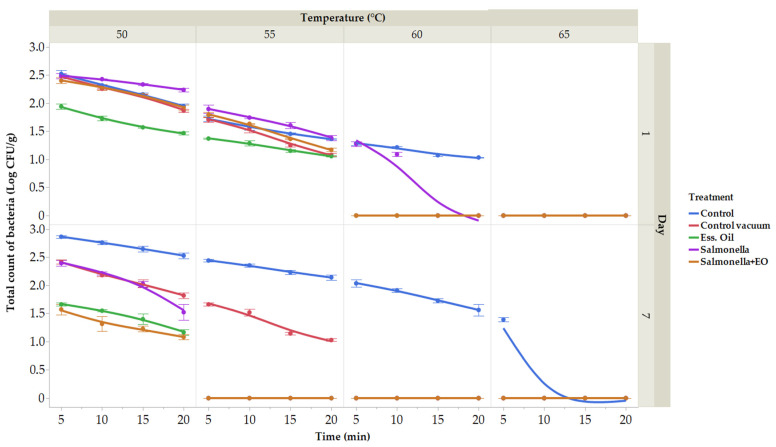
The outcomes of the total bacterial count. Total bacterial count treated at temperatures ranging between 50 and 65 °C for durations of 5 to 20 min. (expressed in log CFU/g) on the first day. Data are the mean (±SD) of 3 samples. Control: Fresh carrot sample was treated at 50–65 °C for 5 to 25 min after being packed in polyethylene bags and kept at 4 °C. Control vacuum: Fresh carrot sample was treated at 50–65 °C for 5 to 25 min after being vacuum-packed in polyethylene bags and kept at 4 °C. EO: vacuum-packed fresh carrot treated with 1% lime EO was kept at 4 °C and treated for 5–25 min at 50–65 °C. *Salmonella*: vacuum-packed fresh carrot treated with *S. enterica* was kept at 4 °C and treated for 5–25 min at 50–65 °C. *Salmonella* + EO: vacuum-packed fresh carrot treated with *S. enterica* and 1% lime EO was kept at 4 °C and treated for 5–25 min at 50–65 °C.

**Figure 5 plants-13-00524-f005:**
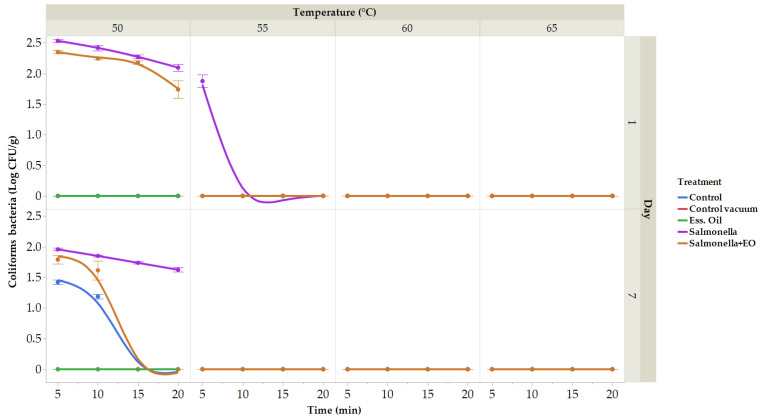
The outcomes of the coliform bacteria. Coliform bacteria treated at temperatures ranging between 50 and 65 °C for durations of 5 to 20 min (expressed in log CFU/g) on the seven day. Data are the mean (±SD) of 3 samples. Control: Fresh carrot sample was treated at 50–65 °C for 5 to 25 min after being packed in polyethylene bags and kept at 4 °C. Control vacuum: Fresh carrot sample was treated at 50–65 °C for 5 to 25 min after being vacuum-packed in polyethylene bags and kept at 4 °C. EO: vacuum-packed fresh carrot treated with 1% lime EO was kept at 4 °C and treated for 5–25 min at 50–65 °C. *Salmonella*: vacuum-packed fresh carrot treated with *S. enterica* was kept at 4 °C and treated for 5–25 min at 50–65 °C. *Salmonella* + EO: vacuum-packed fresh carrot treated with *S. enterica* and 1% lime EO was kept at 4 °C and treated for 5–25 min at 50–65 °C.

**Figure 6 plants-13-00524-f006:**
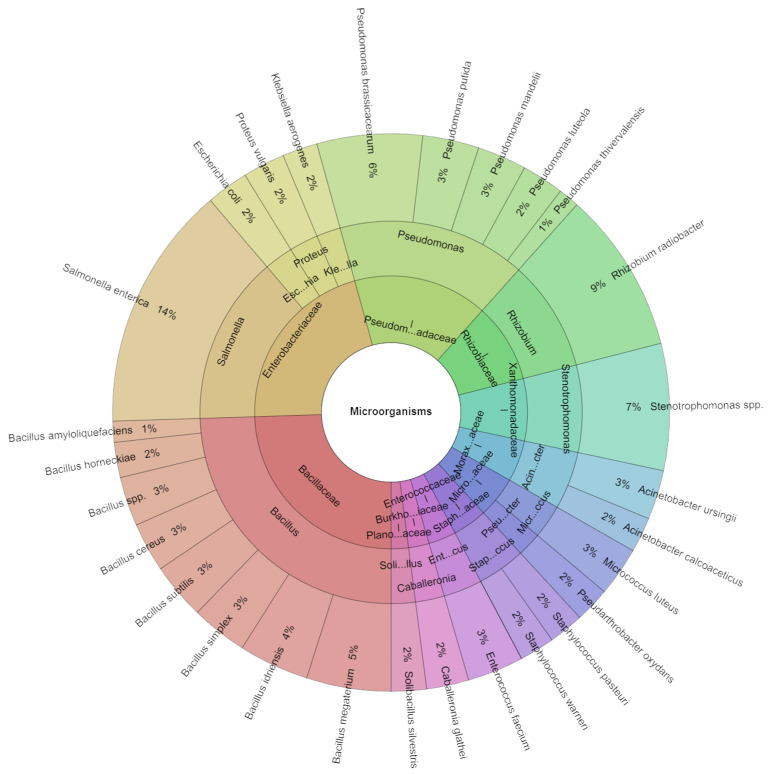
Krona chart: Isolated species of carrot sous vide samples of bacteria in % at day 1.

**Figure 7 plants-13-00524-f007:**
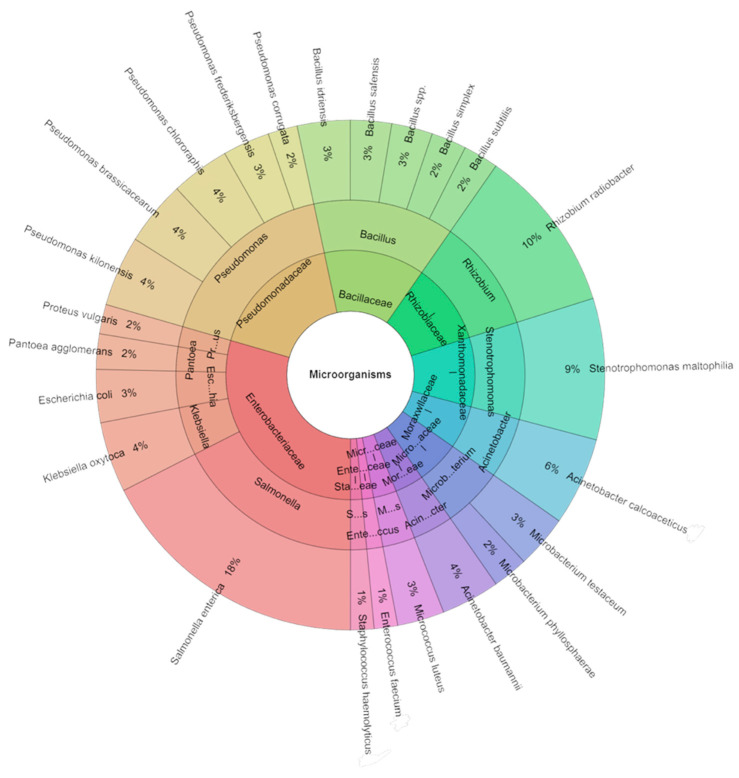
Krona chart: Isolated species of carrots sous vide samples of bacteria in % at day 7.

**Table 1 plants-13-00524-t001:** Chemical composition of *Citrus limon* green essential oil.

No	RI (Lit) ^a^	RI (Cal) ^b^	Compound ^c^	% ^d^
			**non-Terpenic Compounds**	**0.1**
			** *alkanes* **	***tr* ^e^**
1	1400	1400	n-tetradecane	tr
			** *alcohols* **	** *tr* **
2	1068	1077	n-octanol	tr
3	1169	1176	n-nonanol	tr
			** *aldehydes* **	** *0.1* **
4	998	1004	n-octanal	tr
5	1100	1106	n-nonanal	0.1
6	1201	1206	n-decanal	tr
7	1306	1309	n-undecanal	tr
			** *esters* **	** *tr* **
8	1312	1311	n-nonyl acetate	tr
			**monoterpenes**	**97.8**
			** *monoterpene hydrocarbons* **	** *92.4* **
9	930	927	α-thujene	0.5
10	939	935	α-pinene	2.6
11	954	951	camphene	0.1
12	975	974	sabinene	2.4
13	979	980	β-pinene	12.6
14	990	989	β-myrcene	1.9
15	1002	1007	α-phellandrene	tr
16	1017	1019	α-terpinene	0.2
17	1024	1028	p-cymene	0.5
18	1029	1039	limonene	60.7
19	1050	1053	(*E*)-β-ocimene	0.1
20	1059	1066	γ-terpinene	10.3
21	1088	1088	α-terpinolene	0.5
			** *oxygenated monoterpenes* **	** *5.4* **
			*monoterpene alcohols*	*0.9*
22	1096	1100	linalool	0.2
23	1177	1183	terpinen-4-ol	0.1
24	1188	1195	α-terpineol	0.3
25	1229	1224	nerol	0.1
26	1252	1249	geraniol	0.2
			*monoterpene aldehydes*	*3.4*
27	1153	1158	citronellal	tr
28	1238	1238	neral	1.2
29	1267	1267	geranial	2.2
30	1271	1274	perilla aldehyde	tr
			*monoterpene ketones*	*tr*
31	1146	1151	camphor	tr
32	1243	1243	carvone	tr
			*monoterpene epoxides*	*tr*
33	1136	1137	cis-limonene oxide	tr
34	1142	1142	trans-limonene oxide	tr
			*monoterpene esters*	*1.1*
35	1352	1352	citronellyl acetate	tr
36	1361	1361	neryl acetate	0.6
37	1379	1380	geranyl acetate	0.5
			**sesquiterpenes**	** *1.8* **
			*sesquiterpene hydrocarbons*	*1.8*
38	1419	1420	(*E*)-caryophyllene	0.3
39	1434	1434	α-trans-bergamotene	0.6
40	1442	1438	(*Z*)-β-farnesene	tr
41	1456	1454	(*E*)-β-farnesene	tr
42	1496	1492	valencene	tr
43	1500	1495	bicyclogermacrene	tr
44	1505	1506	β-bisabolene	0.9
45	1507	1530	(*Z*)-α-bisabolene	tr
			**total**	**99.7**

^a^ Literature values of retention indices on HP-5MS column; ^b^ Calculated values of retention indices on HP-5MS column; ^c^ identified compounds; ^d^ percentage amounts of identified compounds; ^e^ tr-compounds identified in amounts less than 0.1%.

**Table 2 plants-13-00524-t002:** Disc diffusion method antimicrobial activity in mm.

Microorganism	Inhibition Zone	ATB *
G^+^		
*Bacillus cereus* CCM 2010	5.67 ± 0.58 ^bcd^	27.76 ± 0.47
*Micrococcus luteus* CCM 732	7.67 ± 0.58 ^a^	29.33 ± 0.94
*Staphylococcus aureus* CCM 3953	6.33 ± 0.58 ^abc^	30.33 ± 0.48
G^−^		
*Escherichia coli* CCM 3953	4.67 ± 0.58 ^cd^	29.67 ± 0.48
*Vibrio parahaemolyticus* CCM 5937	5.00 ± 1.00 ^bcd^	30.33 ± 1.25
*Yersinia enterocolitica* CCM 7204T	4.67 ± 0.58 ^cd^	28.67 ± 0.48
Yeasts		
*Candida albicans* CCM 8186	6.33 ± 0.58 ^abc^	29.33 ± 0.48
*Candida glabrata* CCM 8270	6.00 ± 1.00 ^abcd^	28.67 ± 0.47
*Candida krusei* CCM 8271	6.67 ± 0.58 ^ab^	28.00 ± 0.82
*Candida tropicalis* CCM 8223	6.33 ± 0.58 ^abc^	29.67 ± 0.94
Biofilm forming bacteria (BFB)		
*Salmonella enterica*	4.33 ± 0.58 ^d^	30.33 ± 0.48

Data are the mean (±SD) of 3 measurements. Different letters in the second column refer to significant differences (Tukey, *p* ≤ 0.05). ATB = Antibiotics; * = the data show no significant differences.

**Table 3 plants-13-00524-t003:** Minimal inhibition concentration and minimal biofilm inhibition concentration of CLEO in mg/mL.

Microorganism	MIC_50_	MIC_90_
G^+^		
*Bacillus cereus* CCM 2010	3.28 ± 0.17 ^d^	3.62 ± 0.17 ^d^
*Micrococcus luteus* CCM 732	2.33 ± 0.39 ^de^	2.52 ± 0.43 ^de^
*Staphylococcus aureus* CCM 3953	6.19 ± 0.25 ^c^	6.37 ± 0.28 ^c^
G^−^		
*Escherichia coli* CCM 3953	22.61 ± 1.05 ^a^	22.80 ± 1.10 ^a^
*Vibrio parahaemolyticus* CCM 5937	6.23 ± 0.34 ^c^	6.35 ± 0.17 ^c^
*Yersinia enterocolitica* CCM 7204T	12.36 ± 0.52 ^b^	12.58 ± 0.54 ^b^
Yeasts		
*Candida albicans* CCM 8186	23.33 ± 0.51 ^a^	24.03 ± 0.79 ^a^
*Candida glabrata* CCM 8270	3.33 ± 0.10 ^d^	3.48 ± 0.06 ^d^
*Candida krusei* CCM 8271	12.09 ± 0.41 ^b^	12.28 ± 0.35 ^b^
*Candida tropicalis* CCM 8223	6.30 ± 0.16 ^c^	6.41 ± 0.16 ^c^
Biofilm forming bacteria (BFB)		
*Salmonella enterica*	1.37 ± 0.42 ^e^	1.47 ± 0.50 ^e^

Data are the mean (± SD) of 3 samples. Different letters in each column refer to significant differences (Tukey, *p* ≤ 0.05).

**Table 4 plants-13-00524-t004:** In situ analysis of the antimicrobial activity (inhibition of microbial growth (%)) of the vapour phase of CLEO in apple, carrot, and kohlrabi.

Food Model	Microorganisms	Concentration of EO (μg/L)			
Apple		62.5	125	250	500
G^+^	*Bacillus cereus*	65.17 ± 2.31 ^b^	23.88 ± 2.40 ^f^	43.62 ± 2.00 ^c^	34.72 ± 3.13 ^d^
	*Micrococcus luteus*	−13.46 ± 0.53 ^h^	−33.17 ± 1.40 ^g^	33.12 ± 2.10 ^d^	25.00 ± 1.60 ^e^
	*Staphylococcus aureus*	6.26 ± 1.14 ^g^	34.76 ± 2.03 ^e^	55.68 ± 2.02 ^b^	65.41 ± 2.07 ^b^
G^−^	*Escherichia coli*	57.32 ± 2.21 ^c^	46.35 ± 2.67 ^d^	12.07 ± 1.04 ^e^	35.54 ± 1.54 ^d^
	*Vibrio parahaemolyticus*	44.08 ± 1.47 ^d^	23.31 ± 2.29 ^f^	34.80 ± 3.06 ^d^	55.91 ± 2.60 ^c^
	*Yersinia enterocolitica*	16.71 ± 2.30 ^f^	23.00 ± 1.17 ^f^	−56.40 ± 2.26 ^f^	74.63 ± 1.59 ^a^
Yeast	*Candida albicans*	46.57 ± 2.11 ^d^	75.40 ± 2.83 ^b^	45.44 ± 3.09 ^c^	35.19 ± 2.52 ^d^
	*Candida glabrata*	95.54 ± 1.59 ^a^	64.91 ± 2.62 ^c^	44.20 ± 1.98 ^c^	26.11 ± 2.24 ^e^
	*Candida krusei*	94.81 ± 3.59 ^a^	64.17 ± 1.43 ^c^	45.17 ± 3.27 ^c^	26.78 ± 1.63 ^e^
	*Candida tropicalis*	33.77 ± 1.64 ^e^	95.67 ± 1.85 ^a^	75.17 ± 2.43 ^a^	24.47 ± 3.18 ^e^
BFB	*Salmonella enterica*	55.39 ± 3.59 ^c^	45.17 ± 2.07 ^d^	34.21 ± 2.07 ^d^	15.11 ± 1.62 ^f^
**Carrot**					
G^+^	*Bacillus cereus*	74.04 ± 1.66 ^b^	43.61 ± 1.51 ^de^	24.36 ± 1.69 ^e^	−23.89 ± 2.37 ^e^
	*Micrococcus luteus*	−34.47 ± 1.51 ^h^	35.50 ± 3.11 ^e^	25.63 ± 2.76 ^e^	−24.51 ± 2.42 ^ef^
	*Staphylococcus aureus*	34.80 ± 2.76 ^d^	44.93 ± 2.26 ^d^	63.59 ± 1.80 ^b^	75.90 ± 2.67 ^b^
G^−^	*Escherichia coli*	63.88 ± 1.05 ^c^	−28.57 ± 6.11 ^g^	−15.87 ± 1.76 ^f^	75.02 ± 3.08 ^b^
	*Vibrio parahaemolyticus*	25.82 ± 2.02 ^e^	43.83 ± 1.96 ^de^	86.34 ± 3.09 ^a^	−31.91 ± 3.55 ^f^
	*Yersinia enterocolitica*	16.72 ± 1.95 ^f^	75.43 ± 2.93 ^a^	25.16 ± 3.30 ^e^	93.34 ± 1.62 ^a^
Yeast	*Candida albicans*	26.23 ± 2.91 ^e^	45.10 ± 3.12 ^d^	84.85 ± 1.85 ^a^	33.81 ± 3.51 ^c^
	*Candida glabrata*	94.44 ± 1.45 ^a^	64.85 ± 2.67 ^b^	44.36 ± 2.35 ^c^	26.82 ± 2.77 ^cd^
	*Candida krusei*	−14.18 ± 1.44 ^g^	14.95 ± 2.52 ^f^	35.47 ± 2.51 ^d^	26.51 ± 2.51 ^cd^
	*Candida tropicalis*	94.41 ± 2.10 ^a^	63.92 ± 3.09 ^b^	44.99 ± 1.24 ^c^	25.77 ± 2.95 ^d^
BFB	*Salmonella enterica*	65.98 ± 3.94 ^c^	54.51 ± 2.09 ^c^	45.50 ± 2.49 ^c^	25.73 ± 1.17 ^d^
**Kohlrabi**					
G^+^	*Bacillus cereus*	54.57 ± 2.93 ^c^	36.12 ± 1.28 ^e^	−3.93 ± 0.30 ^e^	−13.52 ± 1.75 ^g^
	*Micrococcus luteus*	54.95 ± 2.59 ^c^	−14.21 ± 1.67 ^h^	35.51 ± 2.61 ^c^	13.62 ± 2.55 ^f^
	*Staphylococcus aureus*	54.51 ± 2.03 ^c^	75.29 ± 2.74 ^a^	24.46 ± 2.61 ^d^	85.53 ± 2.08 ^b^
G^−^	*Escherichia coli*	19.24 ± 2.50 ^e^	55.51 ± 1.56 ^c^	45.62 ± 2.48 ^b^	−15.14 ± 2.02 ^g^
	*Vibrio parahaemolyticus*	36.91 ± 3.26 ^d^	54.56 ± 2.47 ^c^	−14.42 ± 2.07 ^f^	84.19 ± 3.07 ^b^
	*Yersinia enterocolitica*	35.77 ± 3.37 ^d^	6.37 ± 2.05 ^g^	55.17 ± 2.43 ^a^	93.48 ± 2.34 ^a^
Yeast	*Candida albicans*	76.63 ± 4.29 ^b^	56.47 ± 3.26 ^c^	44.52 ± 3.20 ^b^	13.76 ± 2.79 ^f^
	*Candida glabrata*	33.23 ± 2.60 ^d^	46.48 ± 2.45 ^d^	55.18 ± 1.97 ^a^	94.41 ± 1.19 ^a^
	*Candida krusei*	97.56 ± 2.53 ^a^	64.74 ± 2.71 ^b^	46.49 ± 2.18 ^b^	25.87 ± 1.63 ^e^
	*Candida tropicalis*	15.76 ± 1.06 ^e^	24.65 ± 1.55 ^f^	35.18 ± 3.57 ^c^	46.07 ± 3.39 ^c^
BFB	*Salmonella enterica*	76.32 ± 3.30 ^b^	65.48 ± 2.12 ^b^	53.48 ± 1.04 ^a^	36.82 ± 1.86 ^d^

Data are the mean (± SD) of 3 samples. Different letters in each column (for each type: apple, carrot, and kohlrabi) refer to significant differences (Tukey, *p* ≤ 0.05).

**Table 5 plants-13-00524-t005:** Insecticidal activity of CLEO against *Harmonia axyridis*.

Concentration (%)	Number of Living Individuals	Number of Dead Individuals	Insecticidal Activity (%)
100	0	100	100.00 ± 0.00
50	10	90	90.00 ± 0.00
25	25	75	75.00 ± 0.00
12.5	50	50	50.00 ± 0.00
6.25	75	25	25.00 ± 0.00
3.125	90	10	10.00 ± 0.00
Control group	100	0	0.00 ± 0.00

## Data Availability

Data will be made available upon request.

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
