# Peer review of "Citrus limon Essential Oil: Chemical Composition and Selected Biological Properties Focusing on the Antimicrobial (In Vitro, In Situ), Antibiofilm, Insecticidal Activity and Preservative Effect against Salmonella enterica Inoculated in Carrot"

_plants, 2024, doi:10.3390/plants13040524_

Round 1
Reviewer 1 Report
Comments and Suggestions for Authors
Author Response
Reviewer #1
The Authors are very grateful to the Reviewer for their valuable comments. We want to thank the Reviewer for the time devoted to point out constructive and important comments to improve our paper.
Point 1: 141-143 delete this text, or add substantive text.
Response: This text has been removed.
Point 2: What was the raw material for obtaining CLEO: “immature fruits of C. limon” (l.157); “freshly harvested petals of the green Citrus limon” (l.573) or “fresh pericarp powder of C. limon”(l.828)?
Response: The essential oil was obtained by cold pressing the fresh pericarp of green lemons. The raw material was consolidated.
Point 3: 163-164 The share of non-terpene compounds is 0.1% (Table 1), is it worth mentioning them? if so, it should be noted that they occur in trace amounts.
Response: According to your comment we have rewritten the sentence in Lines 163-164.
Point 4: No explanations of the abbreviations ATB, a,b,c,d. Does the ATB column describe the effects of known antibiotics (as described in section 3.3.2)? There is no reference to these studies in the text. Does the abcd label mean less statistical significance? Please complete the description under tables 2,3,4 or in chapter 3.8.
Response: The letters are explained in the footnotes. Different letters in each column refer to significant differences. Explanations of the abbreviations ATB was added.
Point 5: It was written “EO had a stronger antibacterial impact against E. coli than it did against Salmonella”, but in 216 line “most resistant was E. coli with MIC 50 of 22.61 mg/mL and MIC 90 of 22.80 mg/mL”. Table 3 shows that the MIC50 and MIC90 values for Salmonella enterica are 1.37 and 1.47 mg/ml, respectively. Further it was written "most sensitive to the tested CLEO was V. parahaemolyticus", according to the data in Table 3, the lowest MIC50 and MIC 90 values are for S.enterica and M.luteus. Please explain.
Response: The sentence was corrected.
Point 6: Explanations "abcde" as in table 2 The line before Candida…. described as "G+" instead of "Yeasts".
Response: The table has been corrected. The letters are explained in the footnotes. Different letters in each column refer to significant differences. Explanations of the abbreviations ATB was added.
Point 7: Explanations all "abcdefgh" as in table 2.
Response: The letters are explained in the footnotes. Different letters in each column refer to significant differences. Explanations of the abbreviations ATB was added.
Point 9: The description under the figure is inconsistent with the figures, where it reads SEPC/SEP/SES+Day. In the next figure, the description is completely different: Day+PCSE/SECLG/SECLS Please standardize.
Response: It was corrected.
Point 10: Check the correctness of the description in the text with the data in Figure 2. Do the SEP figures refer to tests on glass? For example, in Fig.3D - you can see clear differences in SEP and SES spectra, while the text describes "except on day 9 when the spectra exhibited considerable similarity".
Response: It was corrected.
Point 11: “In our study S. enterica antibiofilm activity was evaluated” should be "Citrus limon"
Response: It was corrected.
Point 12: In the "Insecticidal Activity (%)" column, do the values ± 0.00 refer to SD (no explanation) and is it 0.00 in every case?
Response: In each case, the insecticidal activity with SD was 0±00. Thus, the number of killed and unkilled individuals was the same in all cases.
Point 13: The lowest concentration is 3.125% - in the text 3.15%. Incomprehensible sentences
Response: It was corrected.
Point 14: Please check the unit of log CFU/ml, log CFU, or log CFU/g
Response: It was checked.
Point 15: What is the difference between "Control" and "Control Vacuum" group, the description is the same.
Response: It was corrected. Control is without vacuum, just packed. Control Vacuum is vacuum/packed.
Point 16: 500ml – it should be 500µl
Response: It was corrected.
Point 17: 250 µL eppendorf flask is too small for 300 µL of distilled water and 900 µL of ethanol
Response: 200 µL of stock solution was taken into 250 µL eppendorf flasks and mixed with HCCA matrix.
Point 18: Please take into account any changes in the text also in your conclusions.
Response: It was corrected.
Point 19: Incorrect literature entry.
Response: It was corrected.
Reviewer 2 Report
Comments and Suggestions for Authors
Microbiological Analyzes - I suggest the source
I suggest a better resolution in the figures
Author Response
Reviewer #2
The Authors are very grateful to the Reviewer for their valuable comments. We want to thank the Reviewer for the time devoted to point out constructive and important comments to improve our paper.
Point 1: Microbiological Analyzes - I suggest the source.
Response: The references was added.
Point 2: I suggest a better resolution in the figures.
Response: It was corrected.
Reviewer 3 Report
Comments and Suggestions for Authors
Dear Editor and Authors,
The revised article discussed the qualitative and quantitative composition of Citrus limon essential and its microbial and insecticidal activity. The essential oil of common lemon has been studied many times and it is very well known. The manuscript is very extensive, consisting of 29 pages. The article was correctly designed. However, many methodological mistakes were appeared. Language of this article should be carefully revised.
Title: The word “action” in the title should be replaced with “activity”
In three different parts of this article you mention different materials for your investigation e.g. line 157 “essential oil obtained grom immature fruits of. C. limon, line: 573 “freshly harvested petals” (Materials & Methods) and line:827-828 “fresh pericarp powder of C. limon”. Please carefully describe what was the plant material for this research.
Line 148: Would you mind check if there is an identification mistake of (Z)-caryophyllene. There is a large difference between retention indices RI (lit)=1408 and RI (cal) 1420. I think it might be (E)-β-caryophyllene RI lit 1417-1415 on your GC-MS column.
Line 144-177 (Chemical composition of Citrus limon essential oil) The chemical composition C. limon essential oil has been the subject of many scientific investigations. Please explain why you did not compare your research with data published previously by other teams. Revised manuscript contains only information about limonene identified in other essential oils. The discussion on the qualitative and quantitative composition should be based on several publications. It this will make the discussion easier which ingredients are responsible for microbiological activity of this essential oil.
There is only two methods of essential oil production: hydrodistillation (steam distillation) of plant material and extrusion of pericarp of citrus fruits. Please explain what essential oil isolation methods were mentioned in line 173.
In lines 186-191 you refer the microbial activity of C. limon peel extracts. Neither activity of plants extracts cannot be compared to the similar activity of essential oils produced form the same plant material. It is caused by their significant differences in the composition of biological active constituents. Please explain why you described microbial activity of extracts in a manuscript relating essential oils.
Line 328: The word “heightened” should be replaced with “increased”
Line 334: The word “utilized” should be replaced with “used”
Line 352: The word “validate” should be replaced with “confirms”
Lines 350-360: In this fragment you refer your results to activity of some extracts published in scientific papers. The described data concerning C. limon essential oil as well as extracts. As I mentioned before the data relating essential oils and extracts should be described separately. Please compere your results to those data which described essential oils activity.
Line 572-576 - Please correct the part regarding materials and methods. Firstly as I mentioned before there is three different plant materials are described in different parts of this article. Secondly essential oil cannot be produced by means of cold-press extractions. Macerate is obtained, using this method, and it will have different composition and activity than essential oil.
Line 572: The word “utilized” should be replaced with “used”
Line 574: The word “emanated” should be replaced with “isolated”
Line 575: The word “diligently” should be replaced with “immediately”
Line 578: The word “meticulously” should be replaced with “carefully”
Line 597: The word “executed” should be replaced with “determined” or “identified”.
I would reconsider publications of this manuscript after major revision.
Best regards

Comments regarding quality of language have been attached above.
Author Response
Reviewer #3
The Authors are very grateful to the Reviewer for their valuable comments. We want to thank the Reviewer for the time devoted to point out constructive and important comments to improve our paper.
Point 1: Title: The word “action” in the title should be replaced with “activity”.
Response: It was changed.
Point 2: In three different parts of this article you mention different materials for your investigation e.g. line 157 “essential oil obtained grom immature fruits of. C. limon, line: 573 “freshly harvested petals” (Materials & Methods) and line:827-828 “fresh pericarp powder of C. limon”. Please carefully describe what was the plant material for this research.
Response: It was corrected.
Point 3: Line 148: Would you mind check if there is an identification mistake of (Z) caryophyllene. There is a large difference between retention indices RI (lit)=1408 and RI (cal) 1420. I think it might be (E)-β- caryophyllene RI lit 1417-1415 on your GC-MS column.
Response: We would like to thank the reviewer for noticing the mistake. After reviewing the original data, we are consistent with the comment that compound identified with retention indices of 1420 is (E)-caryophyllene.
Point 4: Line 144-177 (Chemical composition of Citrus limon essential oil) The chemical composition C. limon essential oil has been the subject of many scientific investigations. Please explain why you did not teams. Revised manuscript contains only information about limonene identified in other essential oils. The discussion on the qualitative and quantitative composition should be based on several publications. It this will make the discussion easier which ingredients are responsible for microbiological activity of this essential oil.
Response: Considering that the concentration of limonene of the sample tested in this study (and in all other species) is identified in significantly high amount compared to the other abundant components it represents a marker of this essential oil (which is discussed in comparison with the published data). Additionally, essential oils express biological activity as a mixture with the elements of the additive, antagonistic, or synergistic effects of the compounds presented in minor amounts and/or different concentrations of the major components. Considering, this study wasn’t structured to assess the antimicrobial effects of the compounds presented in major or minor amount, but to evaluate the potential use of the studied mixture as a food preservative agent.
Point 5: There is only two methods of essential oil production: hydrodistillation (steam distillation) of plant material and extrusion of pericarp of citrus fruits. Please explain what essential oil isolation methods were mentioned in line 173.
Response: The essential oil was commercially purchased from Hanus s.r.o. The manufacturer describes the production of EO as cold pressing of fresh pericarp of green lemons. It was corrected. Dear reviewer, we do not agree with the comment that there are only two methods used for essential oil production. Besides the two mentioned methods, there are also cold pressing, microwave 'dry' distillation, salt-assisted extraction, ultrasound-assisted extraction, enzymes-assisted extraction, distillation by using Clevenger apparatus etc. Please find below some of the already published papers on the subject:
https://doi.org/10.1002/ffj.1829
https://doi.org/10.1080/0972060X.2021.1925596
https://doi.org/10.1088/1757-899X/1122/1/012108
https://doi.org/10.1016/j.cep.2021.108726
https://doi.org/10.1016/S1572-5995(00)80014-9
Point 6: In lines 186-191 you refer the microbial activity of C. limon peel extracts. Neither activity of plants extracts cannot be compared to the similar activity of essential oils produced form the same plant material. It is caused by their significant differences in the composition of biological active constituents. Please explain why you described microbial activity of extracts in a manuscript relating essential oils.
Response: It was replaced to another study. The sentence was replaced with study of essential oil: Yazgan, H.; Özoğul, Y.; Küley, E. Antimicrobial Influence of Nanoemulsified Lemon Essential Oil and Pure Lemon Essential Oil on Food-Borne Pathogens and Fish Spoilage Bacteria. Int J Food Microb 2019, 306, 108266, doi:10.1016/j.ijfoodmicro.2019.108266.
Point 7: Line 328: The word “heightened” should be replaced with “increased”
Response: It was corrected.
Point 8: Line 334: The word “utilized” should be replaced with “used”
Response: It was corrected.
Point 9: Line 352: The word “validate” should be replaced with “confirms”
Response: It was corrected.
Point 10: In this fragment you refer your results to activity of some extracts published in scientific papers. The described data concerning C. limon essential oil as well as extracts. As I mentioned before the data relating essential oils and extracts should be described separately. Please compere your results to those data which described essential oils activity.
Response: This fragment was redescribed and replaced with Citrus EO studies.
Point 11: Line 572-576 - Please correct the part regarding materials and methods. Firstly, as I mentioned before there is three different plant materials are described in different parts of this article. Secondly essential oil cannot be produced by means of cold press extractions. Macerate is obtained, using this method, and it will have different composition and activity than essential oil.
Response: It was corrected and we list the production of the essential oil as declared by the manufacturer.
Point 12: The word “utilized” should be replaced with “used”
Response: It was corrected.
Point 13: The word “emanated” should be replaced with “isolated”
Response: it was corrected.
Reviewer 4 Report
Comments and Suggestions for Authors
The work Citrus limon Essential Oil: Chemical composition and selected biological properties focusing on antimicrobial activity (in vitro, in situ), antibiofilm, insecticidal activity and preservative activity against Salmonella enterica Inoculated in the Carrot presents very interesting and significant results confirming a wide range of properties essential oil obtained from citrus fruits and the possibilities of its use. The results demonstrate the promising antimicrobial and antibiofilm properties of Citrus Limon essential oil, supporting its possible use in food preservation and spoilage management.
Author Response
Reviewer #4
The Authors are very grateful to the Reviewer for their valuable comments. We want to thank the Reviewer for the time devoted to point out constructive and important comments to improve our paper.
Round 2
Reviewer 3 Report
Comments and Suggestions for Authors
Dear Authors,
I recommend this article for publication.
Best regards